# THE EEG ACTIVATION MAPS IN RECENT WORK ARE UNINTERPRETABLE BY EXPERTS

## ABSTRACT

Recent papers claim to decode object class from EEG recordings of subjects viewing image stimuli from ImageNet and to use that classifier to construct activation maps for the depicted object class that are consistent with neuroscience knowledge. Empirical evaluation of the activation maps by EEG experts calls this claim into question.

## 1 INTRODUCTION

A recent sequence of papers (Spampinato et al., 2017; Kavasidis et al., 2017; Palazzo et al., 2017; 2018; 2020a;b; 2021; 2024) claims to decode object class from EEG recordings of subjects viewing image stimuli from ImageNet (Deng et al., 2009) depicting those classes. These papers all use a single dataset whose collection is reported in Spampinato et al. (2017). That EEG data collection protocol has previously been shown to be flawed and suffer from a confound that confuses stimulus class with stimulus presentation time (Li et al., 2021; Ahmed et al., 2021; 2022; Bharadwaj et al., 2023). When the confound is removed, classification accuracy drops to chance. Palazzo et al. (2021), along with an earlier preprint (Palazzo et al., 2018), use the data from Spampinato et al. (2017) to construct activation maps for the object classes. The activation map for a given class purports to represent the regions of the brain that are involved in processing stimulus images of that class. Palazzo et al. (2018, Fig. 6) contains activation maps for all forty stimulus classes used. Palazzo et al. (2021, Fig. 9) contains activation maps for only eight stimulus classes. Palazzo et al. (2018; 2021) generally claim that the activation maps are consistent with neuroscience knowledge.

> *The obtained results show that the learned brain-visual features lead to improved performance and simultaneously bring deep models more in line with cognitive neuroscience work related to visual perception and attention.*
> (Palazzo et al., 2021, abstract, p. 3833)

> *In particular, this paper demonstrates that a) neural activity data can be used to provide richer supervision to deep learning models, resulting in visual classification and saliency detection methods aligned with human neural data; b) joint artificial intelligence and cognitive neuroscience efforts may lead to uncover neural processes involved in human visual perception by maximizing the similarity of deep models with human neural responses. Indeed, we propose a method to extract visual saliency (and its evolution over time), as well as to localize the cortical region producing such information; and c) there is potential similarity between computational representations and brain processes, providing interesting insights about consistency between biological and deep learning models.*
> (Palazzo et al., 2021, §1¶3, p. 3834)

> *Our approach is a step forward towards providing cognitive neuroscientists with AI-based methodology for understanding neural responses both in space and time, without the need to design experiments with multiple subjects and trials. When highly accurate AI is designed, it will allow cognitive neuroscientists to simulate human responses rather than collect significant amounts of costly data.*
> (Palazzo et al., 2021, §1¶4, p. 3834)

*From these results, some interesting conclusions can be drawn: 1) All visual classes rely heavily on early visual areas including V1 cortex—known to be responsible for early visual processing [26]—and this region is important in all tested time windows; 2) The average activation maps over time clearly show that the process starts in early visual areas and then flows to the frontal regions (responsible of [sic] higher cognitive functions) and temporal regions (responsible for visual categorization [10]); 3) The pattern of activation changes with the visual content; e.g., the "piano" or the "electric guitar" visual class, activates scalp regions closer to auditory cortex (left-most and right-most areas of the scalp), and this is in line with evidence that the sensation of sounds is often associated with sight [70].* (Palazzo et al., 2021, §7.6.1¶2, p. 3845, citations in the original)

*Our work most directly relates to the fields of EEG data classification, computational neuroscience for brain decoding, machine learning guided by brain activity and multimodal learning.* (Palazzo et al., 2021, §2¶1, p. 3834)

*The EEG classification approach proposed in this paper aims to improve the architectural design concepts of [21], [22] by modelling more general spatio-temporal features of neural responses with a goal of supporting cognitive neuroscience studies to improve the interpretability of human neural data in time and space.* (Palazzo et al., 2021, §2¶3, p. 3834, citations in the original)

*While drawing general cognitive neuroscience conclusions from these findings is not the main goal of this work, given also the small scale of the cognitive experiment, we propose an AI-based strategy that seems to produce reliable approximations of brain representations and their corresponding scalp activity, by jointly learning a model that maximizes the correlation between neural activity and visual images.* (Palazzo et al., 2021, §6¶2, p. 3847)

Here we attempt to quantitatively evaluate that claim and demonstrate that this claim cannot be maintained.

## 2 SIGNIFICANCE

Several independent lines of research have refuted a large body of flawed work (Spampinato et al., 2017; Kavasidis et al., 2017; Palazzo et al., 2017; 2018; 2020a;b; 2021; 2024) along completely different axes. Li et al. (2021) demonstrated that the dataset used (Spampinato et al., 2017), and the methods used to collect that dataset, suffer from a temporal confound, correlating stimulus class with experiment timing. Accuracy drops to chance when the confound is removed. Ahmed et al. (2021) demonstrated that this holds even with a much larger dataset. Ahmed et al. (2022) demonstrated that this holds for the additional classifiers used in Palazzo et al. (2018; 2020a;b; 2021). Bharadwaj et al. (2023) demonstrated that this holds even when using supertrials.

Here we progress beyond prior demonstrations that the particular dataset (Spampinato et al., 2017) is confounded. We offer the novel claim the activation maps do not appear to be consistent with neuroscience knowledge, at least as judged by expert neuroscientists. This is significant because claimed validity of the activation maps is used to justify continued use of the confounded dataset and continued use of confounded methods to collect new confounded datasets.

This is further significant for the following reasons:

- Nearly one hundred papers (An & Cho, 2016; Spampinato et al., 2016; Ben Said et al., 2017; Bozal Chaves, 2017; Kavasidis et al., 2017; Palazzo et al., 2017; Parekh et al., 2017; Spampinato et al., 2017; Zhang et al., 2017; Du et al., 2018; Fares et al., 2018; Kumar et al., 2018; Palazzo et al., 2018; Piplani et al., 2018; Tirupattur et al., 2018; Wang et al., 2018; Zhang & Liu, 2018; Zhang et al., 2018; Zhong et al., 2018; Du et al., 2019; Hwang et al., 2019; Jiang et al., 2019; Jiao et al., 2019; Long et al., 2019; Mukherjee et al., 2019; Uys,

2019; Wang et al., 2019; Cudlenco et al., 2020; Fares et al., 2020; Li et al., 2020; Palazzo et al., 2020a;b; Wang et al., 2020; Zheng et al., 2020a;b; Palazzo et al., 2021; Zheng & Chen, 2021; Ma et al., 2021; Mo et al., 2021; Jiang et al., 2021; Lee et al., 2021; Cavazza et al., 2022; Khaleghi et al., 2022; Lee et al., 2022; Mishra et al., 2022; Mishra, 2022; Scharnagl & Groth, 2022; Shimizu & Srinivasan, 2022; Ahmadieh et al., 2023; Bai et al., 2023; Du et al., 2023; Duan et al., 2023; Hasan & A, 2023; Imani et al., 2023; Lan et al., 2023; Lee et al., 2023; Liu et al., 2023; Singh et al., 2023; Song et al., 2023; Wahengbam et al., 2023; Zeng et al., 2023b;a; Fan et al., 2024; Ferrante et al., 2024a;b; Gou et al., 2024; Lei et al., 2024; Liu et al., 2024a;b; Luvsansambuu et al., 2024; Mishra et al., 2024; Mwata-Velu et al., 2024; Ngo et al., 2024; Palazzo et al., 2024; Pan et al., 2024; Qian et al., 2024; Singh et al., 2024; Tang et al., 2024; de la Torre-Ortiz et al., 2024; Yang & Liu, 2024; Ye et al., 2024; Zheng et al., 2024b;a; Zhu et al., 2024; Deng et al., 2025; Fares, 2025; Fu et al., 2025; Lopez et al., 2025; Mehmood et al., 2025; Singh et al., 2025; Xiang et al., 2025) draw flawed conclusions based on the confounded dataset from Spampinato et al. (2017) and datasets suffering from the same confound.

- A number of new datasets have been collected with this same confounded protocol (Gou et al., 2024; Pan et al., 2024; Zhu et al., 2024; Qian et al., 2024; Uys, 2019; Shimizu & Srinivasan, 2022; Liu et al., 2024b; Wang et al., 2019; 2020; Ma et al., 2021; Cudlenco et al., 2020; Zheng et al., 2024b; Cavazza et al., 2022; Luvsansambuu et al., 2024; Liu et al., 2023; Bai et al., 2023; Parekh et al., 2017).
- A number of these have been publicly released and are used by others. For example, Singh et al. (2023), Singh et al. (2024), and Lopez et al. (2025) use the dataset reported in Kumar et al. (2018) and Duan et al. (2023), Singh et al. (2024), and Lopez et al. (2025) use the dataset reported in Ma et al. (2021).
- This is further egregious because Palazzo et al. (2020b; 2024) continue to claim that their dataset (Spampinato et al., 2017), and their results that were obtained with that dataset (Spampinato et al., 2017; Kavasidis et al., 2017; Palazzo et al., 2017; 2018; 2020a;b; 2021; 2024), are valid, despite the refutations in Li et al. (2021), Ahmed et al. (2021; 2022), and Bharadwaj et al. (2023), in part, because of claims I–IV in Palazzo et al. (2018; 2020a;b; 2021).
- This has been used to justify continued publication of a large and growing body of flawed work based on confounded datasets (Cavazza et al., 2022; Khaleghi et al., 2022; Lee et al., 2022; Mishra et al., 2022; Mishra, 2022; Scharnagl & Groth, 2022; Shimizu & Srinivasan, 2022; Ahmadieh et al., 2023; Bai et al., 2023; Du et al., 2023; Duan et al., 2023; Hasan & A, 2023; Imani et al., 2023; Lan et al., 2023; Lee et al., 2023; Liu et al., 2023; Singh et al., 2023; Song et al., 2023; Wahengbam et al., 2023; Zeng et al., 2023b;a; Fan et al., 2024; Ferrante et al., 2024a;b; Gou et al., 2024; Lei et al., 2024; Liu et al., 2024a;b; Luvsansambuu et al., 2024; Mishra et al., 2024; Mwata-Velu et al., 2024; Ngo et al., 2024; Palazzo et al., 2024; Pan et al., 2024; Qian et al., 2024; Singh et al., 2024; Tang et al., 2024; de la Torre-Ortiz et al., 2024; Yang & Liu, 2024; Ye et al., 2024; Zheng et al., 2024b;a; Zhu et al., 2024; Deng et al., 2025; Fares, 2025; Fu et al., 2025; Lopez et al., 2025; Mehmood et al., 2025; Singh et al., 2025; Xiang et al., 2025) even after the confound became known through the work of Li et al. (2021), Ahmed et al. (2021; 2022), and Bharadwaj et al. (2023).

Current machine-learning conferences, and more generally, computer-science conferences and journals, are loathe to publish refutations. Observing this, Schaeffer et al. (2025) proposed that the field of machine-learning establish a "refutations and critiques" track in prominent conferences. While we applaud and support this proposal, the current lack of such a track should not be an impediment to publishing refutations. Scientific journals in other fields have long done so, often resulting in retraction of flawed work. Schaeffer et al. (2025) offer five example pieces of claimed flawed work in machine learning. Each is an individual paper. These pale in comparison to the flaws we uncover here: a systemic flaw of the entire peer review process across an entire field of inquiry, namely classification of stimulus image class from EEG recordings, that affects seventeen datasets and ninety one papers. Moreover, none of the five examples in Schaeffer et al. (2025) are egregious; here the authors of the flawed work continue to argue for its validity despite four refereed refutations and fifty new flawed papers have been published subsequent to these four refereed refutations. This argues for the need to make the community aware of the severity of the issue.

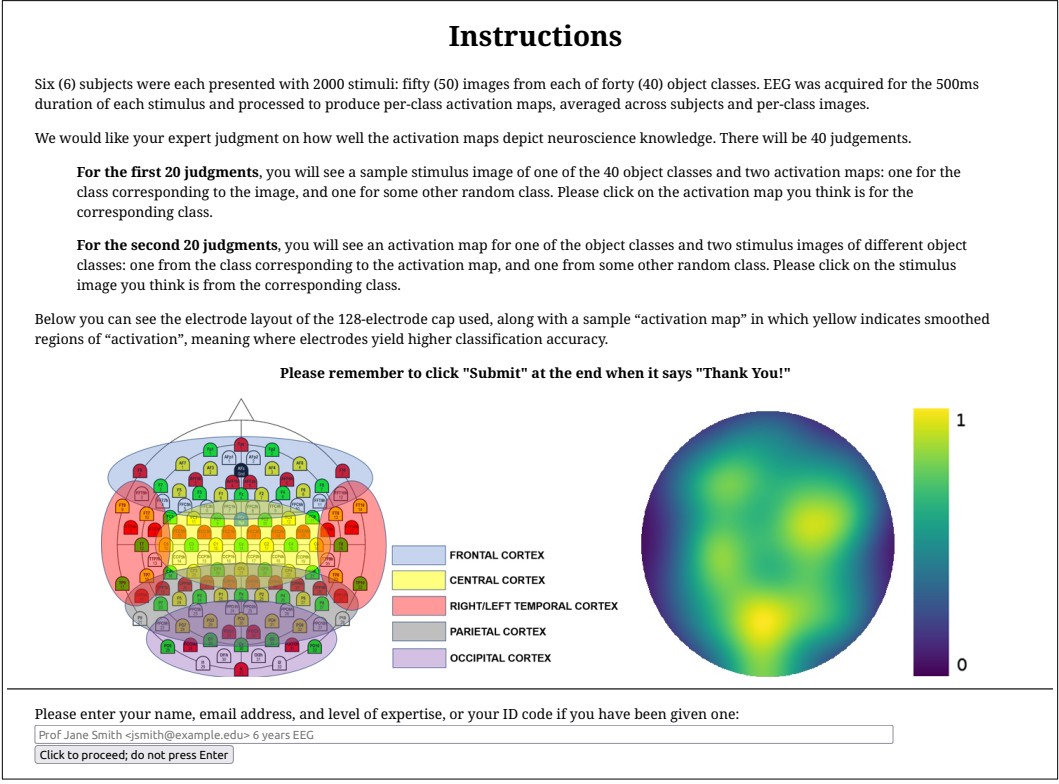

**Instructions**

Six (6) subjects were each presented with 2000 stimuli: fifty (50) images from each of forty (40) object classes. EEG was acquired for the 500ms duration of each stimulus and processed to produce per-class activation maps, averaged across subjects and per-class images.

We would like your expert judgment on how well the activation maps depict neuroscience knowledge. There will be 40 judgements.

**For the first 20 judgments**, you will see a sample stimulus image of one of the 40 object classes and two activation maps: one for the class corresponding to the image, and one for some other random class. Please click on the activation map you think is for the corresponding class.

**For the second 20 judgments**, you will see an activation map for one of the object classes and two stimulus images of different object classes: one from the class corresponding to the activation map, and one from some other random class. Please click on the stimulus image you think is from the corresponding class.

Below you can see the electrode layout of the 128-electrode cap used, along with a sample "activation map" in which yellow indicates smoothed regions of "activation", meaning where electrodes yield higher classification accuracy.

**Please remember to click "Submit" at the end when it says "Thank You!"**

FRONTAL CORTEX
CENTRAL CORTEX
RIGHT/LEFT TEMPORAL CORTEX
PARIETAL CORTEX
OCCIPITAL CORTEX

Please enter your name, email address, and level of expertise, or your ID code if you have been given one:

Prof Jane Smith <jsmith@example.edu> 6 years EEG

Click to proceed; do not press Enter

Figure 1: Protocol: instructions

## 3 METHOD

Eighty ImageNet (Deng et al., 2009) images were selected from the 2000 stimulus images used by Spampinato et al. (2017), exactly two different ones for each class. Two distinct images for each of the forty classes were selected randomly from the fifty stimulus images used by Spampinato et al. (2017) for that class. One image for each of the forty classes was designated as image A, the other as image B. This same set of eighty images and assignment to A *vs.* B was used for all of our experiments.

The forty activation maps presented in Palazzo et al. (2018) were extracted from the pdf of the paper. The activation maps were constructed by Palazzo et al. (2018; 2021) by training a classifier on the EEG response of all 6 subjects to all 2000 trials, measuring the differential classification accuracy when all EEG channels are used *vs.* when one EEG channel is knocked out, doing this for all EEG channels one by one, arranging the differential classification accuracies according to the channel layout of the EEG cap, and rendering after low-pas spatial filtering.

A simple protocol was developed that showed expert EEG consultants two sets of twenty trials after an instructions page (Fig. 1). The first set of twenty trials was designed to assess whether experts could map a stimulus class to an activation map (Fig. 2). This used the A images. The second set of twenty trials was designed to assess whether experts could map an activation map to a stimulus class (Fig. 3). This used the B images.

Each of the forty classes was assigned randomly to either the first set of trials or the second. Each of the forty classes corresponded to the correct answer of exactly one of each of the forty trials, collectively across the two sets of twenty trials.

Each trial in the first set presented one image A along with two activation maps, one that corresponded to the class of the image and one that did not (Fig. 2). Consultants were asked to click on the activation map that corresponded to the class of the image. Only twenty out of the forty images A were used, those whose class corresponded to a class selected for the first set of twenty trials. Each

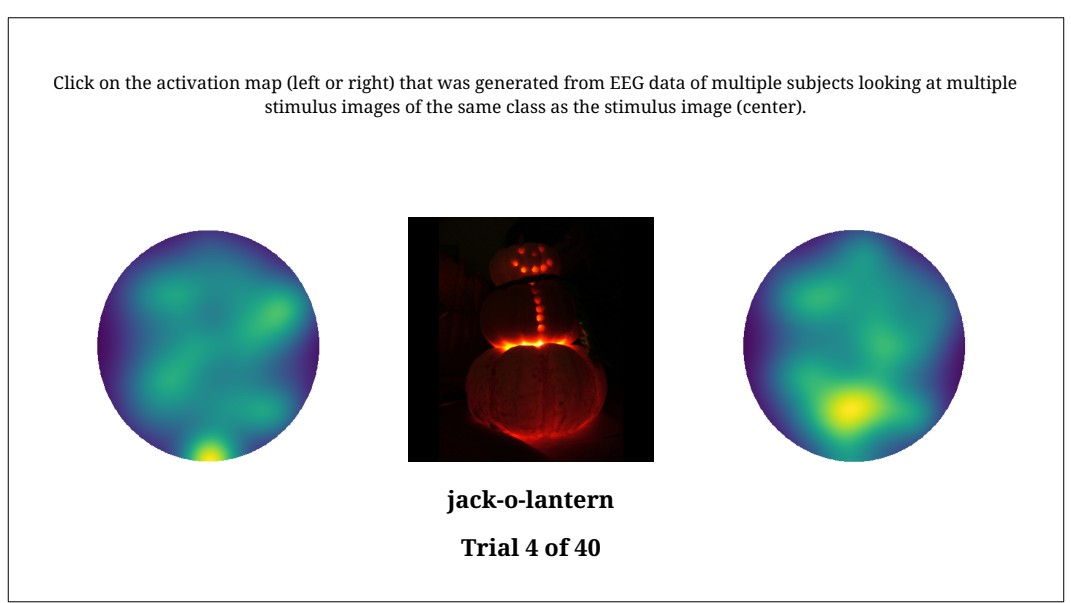

Figure 2: Protocol: format for trials 1–20. This was designed to assess whether experts could map a stimulus class to an activation map.

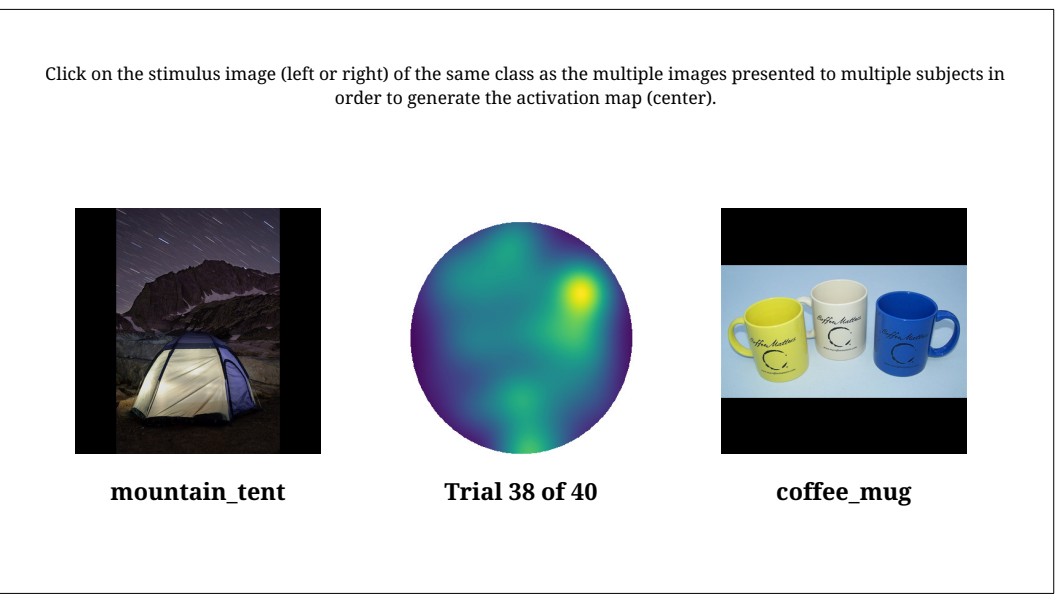

Figure 3: Protocol: format for trials 21–40. This was designed to assess whether experts could map an activation map to a stimulus class.

of these appeared in exactly one trial in the first set. Each of the forty activation maps appeared in exactly one trial in the first set. Each of twenty of these were the correct answer to exactly one trial in the first set. The remaining twenty were not the correct answer to any trial in the first set.

Each trial in the second set presented one activation map along with two images B, one from the class used to construct the activation map and one that was not (Fig. 3). Consultants were asked to click on the image whose class corresponded to the activation map. Only twenty out of the forty activation maps were used, those whose class corresponded to a class selected for the second set of twenty trials. Each of these appeared in exactly one trial in the second set. Each of the forty images B appeared in exactly one trial in the second set. Each of twenty of these were the correct answer to exactly one trial in the second set. The remaining twenty were not the correct answer to any trial in the second set.

The assignment of each of the forty classes to a particular set of trials, the order of trials corresponding to the classes, the choice of distractor images and activation maps, and whether the distractor appeared on the left or the right varied randomly (uniform) across consultant. The system recorded the image(s) and activation map(s) selected for presentation in each trial for each consultant, which appeared on the left *vs.* right, which choice was selected for each trial by each consultant, and the time of each selection.

A dozen consultants, each an expert in EEG analysis, were recruited from local universities and a conference on MEG. These included PhD students, postdocs, researchers, and faculty, with the majority having a PhD. Experience in EEG analysis ranged from 2 to 12 years, with a median of around 7 years.

## 4  RESULTS

The per-consultant number of correct responses was 15, 16, 17, 19, 19, 19, 19, 20, 21, 22, 23, and 27 (out of 40). No consultant performed above chance ($p < 0.005$) as evaluated by a binomial cmf. The mean rate of correct response was $\frac{237}{480} = 49.375\%$ which was also at chance ($p < 0.005$).

## 5  DISCUSSION

The validity of the saliency and activation maps of Palazzo et al. (2018; 2021) has been questioned on the basis of the severely confounded dataset (Li et al., 2021). The saliency maps have further been challenged on theoretical grounds (Li et al., 2021, §5.4). These maps are nonetheless used to justify use of the dataset of Spampinato et al. (2017) on the basis of their purported consistency with known neuroscience.[1] However it appears that any claim that these activation maps reflect neuroscience knowledge cannot be maintained, as not even a single one of our consultants was able to correctly map bidirectionally between object class and activation map with statistically significant above chance accuracy.

## 6  CONCLUSION

The above, together with previous related results (Li et al., 2021; Ahmed et al., 2021; 2022; Bharadwaj et al., 2023), seem to imply that the dataset of Spampinato et al. (2017), and any other dataset that exhibits the confounds discussed in Li et al. (2021), should be avoided, and results based on those datasets must be discounted.

AUTHOR CONTRIBUTIONS

Removed for blind review.

---

[1]Even reviewers seem to accept the validity and importance of the activation maps. *"One major contribution in [3] is the salience map in both image and brain domain. This article only compare [sic] the classification accuracy. It would be interesting to examine whether the salience patterns will become less interoperable."* —Recent journal review; the *"[3]"* refers to Palazzo et al. (2021).

ACKNOWLEDGMENTS

Removed for blind review.

ETHICS STATEMENT

This work debunks nearly one hundred published papers whose results are based on the same con-found: a correlation between stimulus class and temporal drift. This confound has been found in eighteen available EEG datasets. Just as with an inconsistent set of axioms one can prove anything, a confounded dataset can be used to support any claim, even ones that are false or absurd. That is what many recent publications based on this confound do: things like generating high fidelity renderings of images, or even 3D CAD models of objects, from EEG recordings.

A research community, knowingly or unknowingly, has discovered that one can use confounded datasets to churn out a plethora of flawed results without reviewers noticing. They have also dis-covered that one can collect new confounded datasets to churn out even more flawed results without reviewers noticing. The temptation to do this is so strong that the community continues to do so four years after details of the confound were published.

It is conceivable that the flaws in these datasets may be a driving factor behind their frequent reuse. When a dataset is severely confounded, it becomes relatively easy to achieve an extremely high accuracy, which can in turn be used to support sensational claims, and ultimately directs further attention to the dataset. In business, this phenomenon is referred to as "the bad money drives out the good money."

More prominent exposure of these flawed methods and consequent false results will allow resources wasted on continued use of these confounded datasets and flawed methods to be reallocated. The debunked work also causes direct ongoing harm:

- grant proposals can be rejected due to preliminary results not being competitive with results demonstrating falsely-inflated performance based on confounded data or faulty methods;
- manuscripts can be rejected for the same reason;
- grants can be awarded based on false pretenses
- manuscripts can be accepted for the same reason;
- degrees can be awarded for the same reason;
- resources can be wasted attempting to replicate the debunked results;
- resources can be wasted having people read and review flawed papers, and learn flawed methods; and
- because the debunked work relates to brain-computer interfaces—whose primary applica-tion is helping people with disabilities (*e.g.*, paralysis) interact with the world—the harm caused is not merely scientific but also medical, with disproportionate impact on people with disabilities.

REPRODUCIBILITY STATEMENT

All code needed to replicate the experiments presented here will be released upon publication. Anonymized data collected from our trials wil also be released upon publication. Given the col-lected data, the calculations to obtain our results are straightforward and were performed by hand.

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
