# OpenReview forum: "The EEG activation maps in recent work are uninterpretable by experts"
_ICLR.cc/2026/Conference — Submitted to ICLR 2026_

### Official Review · Reviewer_ZyiH · 2025-10-26

**Soundness:** 1
**Presentation:** 1
**Contribution:** 1
**Rating:** 0
**Confidence:** 4

**Summary:**

The paper presents an empirical evaluation of activation maps (generated in another paper) and argues based off the result that the evaluators were unable to map the object class and activation map with above chance accuracy that those activation maps therefore call the validity into question. Furthermore, based off this result the paper argues that this implies that the dataset used in the other paper (Spampinato et al 2017) and other datasets with a similar temporal confound should be avoided and any results drawn from these datasets should be discounted.

**Strengths:**

It is meaningful to highlight limitations in prior work to both help correct misunderstandings that may arise from that prior work and to help push the field forward.

**Weaknesses:**

1. Relevance to call for papers: This work is not aligned with the focus of this conference as defined in the call for papers (https://iclr.cc/Conferences/2026/CallForPapers). While "applications to neuroscience & cognitive science" is a subject area in the call for papers, this is not an application of "feature learning, metric learning, compositional modeling, structured prediction, reinforcement learning, uncertainty quantification and issues regarding large-scale learning and non-convex optimization" or related topics to neuroscience & cognitive science.
2. Relevance to ICLR audience: While highlighting limitations in prior work is valuable, of all the prior work that is being highlighted none of these works were previously published in ICLR. This further supports that this workshop is not the correct venue for this paper, as this limitation will likely be of little interest to the ICLR audience. If there is interest, as the paper itself mentions there are other papers that have previously pointed out limitations to the same body of work this paper focuses on that could be read.
3. Soundness, Claim is not scientifically rigorous: The paper presents extremely limited details about how the "experts" who conducted the empirical evaluation were identified, which limits the strength of the only claim in the paper. Additionally, only summary statistics are presented which prevents the reader from being able to assess the data and draw conclusions on their own about the implication of the empirical evaluation. Furthermore, one of two examples of the empirical evaluation that is shown (Fig 2) raises the question of how the experts would be expected based off their neuroscience knowledge identify which activation map was correct as both overlap with the occipital cortex and the specific localization of jack-o-lanterns is not something that is taught as part of EEG analysis even to experts.
4. Presentation, Motivation is unsupported: The paper boldly claims that "Palazzo et al" generally claims that the "activation maps are consistent with neuroscience knowledge" and then presents 7 quotes from "Palazzo et al" papers. From these 7 Palazzo et al" quotes, only 1 quote presents any support for this claim. The fact that the evidence presented in the paper to support the motivation for highlighting a limitation in the "Palazzo et al" papers is unclear in how it supports the claim is a key weakness.

**Questions:**

It would be valuable if the paper explained:
1. How the experts expertise in EEG analysis was tested before they participated in the empirical evaluation?
2. Whether the experts are in fact experts at object classification activation maps, or if there EEG expertise in in another area?
3. What were the raw results of the empirical analysis? Were there any categories that the experts agreed with the activation maps?
4. The reason why quotes were included in the intro (taking up almost 1 full page of text) when 6 of the 7 quotes do not support the claim they are included to support? Perhaps there is nuance that is not clear from merely reading the quotes.

However, clarifying these points will not chance that the paper is not relevant to the call for papers nor for the ICLR audience. Therefore, I'd recommend the paper be updated to address the above points and submitted at a neuroscience focused venue instead.

---

> ### Author Response · Authors · 2025-11-21
> **Response to ZyiH**
>
> W1: "applications to neuroscience & cognitive science" is a subject
> area in the call for papers
>
> W2: At least one prior flawed paper was published in NeurIPS and two
> flawed manuscripts were submitted to ICLR. (See above for more
> details.)
>
> W3: See responses to above. We would be happy to include the entire
> set of activation maps and images presented to the expert consultants
> in supplementary material. But they are not our own and are already
> public. We obtained the activation maps from a prior publication and
> the released dataset for that prior publication includes the images.
>
> W4: See above.
>
> Q1: A summary of the demographic data is already included in the
> manuscript:
>
> *A dozen consultants, each an expert in EEG analysis, were recruited
>  from local universities and a conference on MEG. These included PhD
>  students, postdocs, researchers, and faculty, with the majority
>  having a PhD. Experience in EEG analysis ranged from 2 to 12 years,
>  with a median of around 7 years.*
>
> We can expand this if the reviewers would find it helpful?
>
> Q2: All of the expert consultants participated enthusiastically and
> reported that they felt they had the expertise to perform the task.
>
> Q3: It was an A/B task with 50% chance level, and we did not observe a
> statistically significant difference from chance performance. This
> implies that the right choice was made about 50% of the time. The
> manuscript includes this information:
>
> *The per-consultant number of correct responses was 15, 16, 17, 19,
>  19, 19, 19, 20, 21, 22, 23, and 27 (out of 40). No consultant
>  performed above chance (p < 0.005) as evaluated by a binomial cmf.
>  The mean rate of correct response was 237/480 = 49.375% which was
>  also at chance (p < 0.005)*
>
> Analysis on a per-question cross-expert level looking for above-chance
> performance on some particular questions would necessarily (given the
> aggregate chance performance) find below-chance performance on other
> questions. This would therefore be difficult to interpret as anything
> other than shot noise. The overall chance performance implies that if
> some particular activation maps were consistent with known
> neuroscience knowledge, this was balanced out by other activation maps
> being inconsistent with known neuroscience knowledge.
>
> Q4: The sole purpose of our manuscript is to remove the support for
> the claims in those quote in the refuted paper.
>
> Note that this is not relevant to a neuroscience focused venue because
> the vast majority of the one hundred flawed papers, and Spampinato et
> al. (2017), Kavasidis et al. (2017), Tirupattur (2018), and Palazzo et
> al. (2018, 2020a, 2020b, 2021 and 2024), were published in CS/AI/ML/CV
> venues, not in neuroscience-focused venues.

---

### Official Review · Reviewer_oV4Z · 2025-10-26

**Soundness:** 2
**Presentation:** 1
**Contribution:** 2
**Rating:** 2
**Confidence:** 3

**Summary:**

This paper addresses a critical issue in EEG-based visual decoding: the lack of expert interpretability of EEG activation maps proposed in a series of high-impact works. It shows that the activation maps—purported to align with neuroscience knowledge—are not interpretable by EEG experts.

**Strengths:**

1. The proposed question is interesting.
2. The references are comprehensive.

**Weaknesses:**

1. Asking humans to interpret EEG spectrograms and map them with true stimuli is an excessively difficult task. Generally, EEG can only capture superimposed general visual features. Hence, the proposed experimental paradigm is not feasible. Meanwhile, this design cannot verify that deep models are unable to extract distinguishable features, as human visual perception is inherently limited.
2. Lack of analysis: only averaged results are reported, no detailed analysis and case-by-case discussions are provided. For example, is there any case where humans can reach high accuracy?
3. The study only demonstrates that the EEG response to a single image cannot be identified by humans, but it cannot prove that the EEG response related to visual perception is uninterpretable.
4. Lack of insight and contribution on how this research can advance the area.

**Questions:**

Could you clarify the definition of "activation maps are consistent with neuroscience knowledge"? Is it a necessary conclusion that the EEG response to a single image cannot be identified by humans?

---

> ### Author Response · Authors · 2025-11-21
> **Response to oV4Z**
>
> W1: We never claim that the proposed task is feasible or that deep
> models are unable to extract distinguishable features. All we claim is
> that the claim in Palazzo et al. (2021) that their activation maps are
> consistent with neuroscience knowledge is unsupported.
>
> W2: We would be happy to add those details if the reviewer thinks they
> would make a difference. All that matters for our refutation claim is
> that we fail to find an effect. More details would not change that.
>
> W3: We never claim that the EEG response related to visual perception
> is uninterpretable. All we claim is that "The EEG activation maps in
> recent work are uninterpretable by experts". That is the title. How
> have we failed to demonstrate this? Note that the activation maps in
> Palazzo et al. (2021) are not the EEG recorded from a single stimulus;
> they are the result of a complicated analysis computed on models
> trained on EEG responses by 6 subjects to 50 images for 40 classes.
> This complicated analysis claims to demonstrate that the model that
> they learn somehow embodies neuroscience knowledge. We simply remove
> support for that claim.
>
> Q: "activation maps are consistent with neuroscience knowledge" is not
> a notion that we introduce. That is a claim of Palazzo et al. (2021)
> that we refute. These activation maps are not produced from a single
> stimulus (see W3 above).
>
> Ethics: This was not a human subjects experiment. It does not yield
> generalizable knowledge. We were asking expert consultants to evaluate
> a scientific claim in a publication.

---

### Official Review · Reviewer_heEM · 2025-10-26

**Soundness:** 1
**Presentation:** 1
**Contribution:** 3
**Rating:** 2
**Confidence:** 5

**Summary:**

The paper mentioned a crucial problem faced by the field of EEG-based visual decoding, which was incurred by the flawed dataset for a previous paper. However, this paper is more like an unfinished comment that needs substantial data analysis.

**Strengths:**

This work pointed out an important issue that is easily faced in the experimental design for brain decoding.

**Weaknesses:**

Hardly any analysis was presented in the article.

**Questions:**

The paper from Li et al. [1] thoroughly analyzed the shortcomings of the work from Spampinato et al. On this basis, what improvements does this article propose?

[1] R. Li et al., "The Perils and Pitfalls of Block Design for EEG Classification Experiments," in IEEE Transactions on Pattern Analysis and Machine Intelligence, vol. 43, no. 1, pp. 316-333, 1 Jan. 2021
[2] C. Spampinato, S. Palazzo, I. Kavasidis, D. Giordano, N. Souly, and M. Shah, “Deep learning human mind for automated visual classification,” in Proc. IEEE Conf. Comput. Vis. Pattern Recognit., 2017, pp. 6809–6817.

---

> ### Author Response · Authors · 2025-11-21
> **Response to heEM**
>
> W: This is a standard analysis of data (collected judgments of expert
> consultants) with the standard statistical significance analysis. It
> refutes a claim in prior work. If the reviewer would specify what
> additional analysis would further clarify the result, we would be
> happy to perform and add that analysis.
>
> Q: As stated above, Palazzo et al. (2020b, 2021, 2024), and over one
> hundred other papers, do not accept that the seriousness of the block
> confound, despite the results of Li et al. (2021). Part of the
> argument to support the use of block-confounded datasets is the
> claimed "neuroscience validity" of the activation maps. Refuting that
> thus contributed to the refutation of a large body of work.

---

### Official Review · Reviewer_kqv8 · 2025-10-30

**Soundness:** 2
**Presentation:** 1
**Contribution:** 2
**Rating:** 2
**Confidence:** 4

**Summary:**

The paper audits published EEG "activation maps" from a prior pipeline by running a blinded expert-judgment study (two forced-choice tasks) to test whether the maps are interpretable to humans. The authors report that experts do not reliably match maps to stimuli and conclude that such maps should not be used to support neuroscience claims about the underlying dataset or model line. The stated contributions are: a focused human-study evaluation of published EEG maps from a prominent prior work, and a critique arguing that these maps, as presented in the literature, do not carry the neuroscientific signal they are often claimed to reflect.

**Strengths:**

1. Clear, concrete target: evaluates what readers actually see in prior publications (published maps), not a moving reimplementation.

2. The overarching question whether these maps are meaningfully interpretable by domain experts is timely and under-tested.

3. Simple, transparent task design that is easy to reproduce.

**Weaknesses:**

1. This is a narrow human-subjects audit with minimal algorithmic or methodological innovation. For an ICLR main track, the CS contribution feels thin; the work reads as a focused methods/ethics evaluation better suited to a special track or journal.

2. The study evaluates maps from a single prior pipeline yet the paper’s narrative gestures toward invalidating a broad literature. The conclusion should be bounded to the artifacts actually tested. This is not a request to "test all papers"; it’s a request to either restrict the claim to the evaluated pipeline, or add a small, diverse sample of other pipelines if the broader claim is essential.

3. There is no positive control using a known good EEG paradigm and/or dataset to verify that the interface/task can detect interpretability when it’s expected to be present. Without this, the null finding could be driven by task design, map rendering, or the natural limits of EEG for fine-grained image classes. Likewise, no negative controls (e.g., randomized/permuted or class-swapped maps) are included to calibrate chance-level behavior.

4. Per-class topographies for natural images are an extremely demanding target for EEG (low SNR, coarse spatial resolution, limited coverage). A failure on this task does not uniquely imply "non-interpretability" of the underlying neural activity; it may reflect a mismatch between the claim tested and what EEG can plausibly support.

5. "EEG expert" is a broad term. The sample mixes backgrounds and seniority, and it’s unclear how many participants have direct experience with cognitive/vision EEG map interpretation versus clinical EEG. This weakens the negative inference.

6. The maps appear heavily smoothed/rasterized rather than recomputed from source. If originals cannot be regenerated, the paper should analyze and justify how extraction/smoothing affects discriminability and discuss robustness to these presentation choices.

7. The tone and structure feel informal for an A* venue (short abstract that under-reports the methods/results, heavy quoting/bullets). The paper would benefit from tighter framing, clearer hypotheses, and more rigorous statistical exposition without going into excessive polemics.

**Questions:**

Will you constrain the central claim to the specific pipeline you evaluated, or add a small sample from other pipelines to justify broader statements?

1. Can you include a positive control (even a compact add-on) using a well-established EEG paradigm to demonstrate that your task/interface can detect interpretability when present?

2. How were "EEG experts" screened for relevance to cognitive/vision EEG topographies (as opposed to clinical EEG)? Would results change if limited to that subgroup?

3. If recomputing maps from source is infeasible for the rebuttal, can you document the exact provenance and quantify how rasterization/smoothing might degrade discriminability (e.g., show side-by-side renderings with/without smoothing or different interpolation)?

4. Can you refine the hypothesis to acknowledge EEG modality limits and rephrase conclusions accordingly (e.g., "these published maps, as rendered and for this task, do not support the claimed interpretability")?

---

> ### Author Response · Authors · 2025-11-21
> **Response to kqv8**
>
> W1: The significance of this work lies in its refutation of prior
> work. That is a contribution, even though it is not algorithmic or
> methodological. Its relevance to the ICLR main track lies in the fact
> that it refutes flawed work published in the ML community.
>
> W2: The claimed neuroscience validity of the activation maps has been
> used by others to argue for the validity of the Perceive dataset and
> the methodology used to collect that dataset. That dataset and
> methodology has been used by over one hundred papers. By refuting the
> validity of the activation maps, support for the validity of the
> Perceive dataset is weakened. If falls under the collective weight of
> all of the other refutations. Once it falls, all of the one hundred
> plus papers fall as well.
>
> W3: Palazzo et al. (2021) claim that the activation maps are
> consistent with neuroscience knowledge. We know of no better way to
> test this claim than to consult expert neuroscientists. The A/B
> paradigm is a standard one in psychology for assessing people's
> judgments in an unbiased way. We make no claim that the task would be
> possible with better activation maps. All we claim is that the fact
> that we get chance performance refutes the claim that these particular
> activation maps are consistent with neuroscience knowledge.
>
> W4: Indeed. All we claim is that Palazzo et al. (2021) did not support
> their claim that their activation maps are consistent with
> neuroscience knowledge.
>
> W5: We have that information and can include it in a revision. That
> said, all had cognitive neuroscience experience; none were purely
> clinical.
>
> W6: We do not have access to the originals and the authors refuse to
> release the originals or the code for regenerating them. We extracted
> therefore used pdfimage to extract the images embedded in the pdfs of
> the published papers.
>
> W7: We regard the manuscript page limit as a maximum, not a target.
> The purpose of this manuscript is to refute the claim that consistency
> of a set of activation maps with known neuroscience supports the
> validity of the confounded dataset, by disproving the antecedent. We
> believe this is accomplished at the current length, but would be happy
> to add additional material should the reviewers feel this would
> strengthen the logic.
>
> Q1: That would show that the activation maps are inconsistent with
> neuroscience knowledge. We are not claiming that. All we are claiming
> is that Palazzo et al. (2021) have not supported their claim that they
> are consistent with neuroscience knowledge. How would doing that
> control further support that claim?
>
> Q2: As stated in the manuscript:
>
> *A dozen consultants, each an expert in EEG analysis, were recruited
>  from local universities and a conference on MEG. These included PhD
>  students, postdocs, researchers, and faculty, with the majority
>  having a PhD. Experience in EEG analysis ranged from 2 to 12 years,
>  with a median of around 7 years.*
>
> We can expand this if the reviewers would find it helpful?
>
> Q3: See W6.
>
> Q4: We will do this.

---

### Author Response · Authors · 2025-11-21
**Relevance and Significance**

To understand this work's significance, consider this brief historical
overview.

Spampinato et al. (2017) introduced a block-designed dataset
("Perceive") and methods that claim to achieve extremely high accuracy
decoding stimulus image class from EEG recordings. This was amplified
by follow on papers (Kavasidis et al. 2017, Palazzo et al. 2018,
2020a, 2020b, 2021), many of which claim to do things like reconstruct
stimulus images from EEG recordings. Further, Tirupattur (2018) does
this with a fresh dataset (Kumar 2018) that has the same block design.

Li et al. (2021) debunked all of the above, demonstrating that the
Perceive dataset suffers from a block confound. EEG exhibits drift,
encoding a clock in the signal. Since Perceive was collected with all
and only stimuli of the same class being temporally adjacent, the
classifier can mistakenly classify the clock/drift instead of the
stimulus-related EEG response. Follow on papers (Ahmed et al. 2021,
2022, Bharadwaj et al. 2023) added novel independent confirmation of
the results of Li et al. (2021).

Despite this, Palazzo et al. (2020b, 2021, 2024) continue to argue
that their dataset is valid. At this point, there are over one hundred
papers that use the Perceive dataset, the Kumar (2018) dataset, or
other datasets that suffer from the same block confound. Many new
datasets have been collected with this same block confound, some of
which are becoming widely used. The vast majority of these were
published after the confound became known (Li et al. 2021). Some of
these are unaware of the confound. Others are aware, but dismiss it,
often based on the argument of Palazzo et al. (2020b, 2021, 2024).

That argument is what this manuscript refutes.

---

> ### Author Response · Authors · 2025-11-21
> **Specific relevance and significance to ICLR and the ML community**
>
> It is important, if not imperative, for the community to publish this
> work. Without it, the community continues to submit and publish more
> flawed work at a growing rate. Fifty new papers papers have been published
> since the flaw was first reported in prominent venues: once in CVPR (Ahmed et al. 2021)
> and three times in TPAMI (Li at al. 2021, Ahmed et al. 2022, Bharadwaj et al 2023).
>
> Some recent flawed work has been published even by the ML community in
> top ML venues, despite awareness of the issue: Liu et al. (2024) in
> NeurIPS collects a new dataset that suffers from the block
> confound. While the authors cite Li et al. (2021) and Ahmed et al
> (2021), they fail to appreciate (or maybe hide the fact) that their
> work is confounded.
>
> Some recent flawed work has even been submitted to ICLR 2025 (and
> apparently resubmitted to ICLR 2026 despite reviewer warnings). It
> appears that even the reviewer pool of ICLR is unaware of the severity
> of the confound.
>
>    https://openreview.net/forum?id=ejVuTFFkl6&noteId=zafmRtlFw1
>
> collects a new dataset that suffers from the block
> confound. While the authors again cite Li et al. (2021), they
> incorrectly claim that their dataset does not suffer from the
> confound. All four of the reviewers point this out. While this
> submission was rejected, three of the reviewers rated it as
> "Soundness: 3: good" and two of the reviewers rated it as
> "Contribution: 3: good".
>
> The apparent resubmission (18265) to ICLR 2026 again cites Li et
> al. (2021) and again incorrectly claims that their dataset does not
> suffer from the confound. Again, three of the four reviewers point out
> that this work suffers from the block confound. And again, two of the
> reviewers rate this as "Soundness: 3: good", one of the reviewers
> rates this as "Contribution: 3: good", and one even rates this as
> "Contribution: 4: excellent" and recommends acceptance.
>
> We have a simple question for the reviewers, area chairs, and program
> chairs: If one cannot publish refutations like this in ICLR, how else
> do you propose we address the fact that there is a large and growing
> body of flawed work being published?

---

### Meta-Review · Area_Chair_BwLR · 2025-12-27

**Summary:**

Reviewers all agree that the work fits poorly for a paper in an ML conference like ICLR given its lack of CS / ML contribution. While the opinion expressed by authors in the paper should make the ML community reflect on its claims regarding neuroscience findings it is clear that ICLR is not the right venue. For these reasons, this paper cannot be endorsed for publication at ICLR.

As an AC I would encourage the authors to push more in the direction as it should help the community progress. Yet, it feels that the format and the venue is not the best fit. I would encourage authors to publish in a journal, also reach out to PIs who publish with wrong claims and write posts on social media to raise awareness.

**Reviewer Concerns:**

- lack of CS/ML contribution
- broad claims from authors do not match with the actual work presented

**Reviewer Scores:**

Reviewer kqv8 2
Reviewer heEM 2
Reviewer ZyiH 0

---

### Decision · Program_Chairs · 2026-01-26

Reject